# Comprehensive Review of the Properties and Modifications of Carbon Fiber-Reinforced Thermoplastic Composites

**DOI:** 10.3390/polym13152474

**Published:** 2021-07-27

**Authors:** Basheer A. Alshammari, Mohammed S. Alsuhybani, Alaa M. Almushaikeh, Bander M. Alotaibi, Asma M. Alenad, Naif B. Alqahtani, Abdullah G. Alharbi

**Affiliations:** 1Material Science Research Institute, King Abdulaziz City for Science and Technology, Riyadh 11442, Saudi Arabia; nqahtani@kacst.edu.sa; 2Nuclear Science Research Institute, King Abdulaziz City for Science and Technology, Riyadh 11442, Saudi Arabia; sohybani@kacst.edu.sa (M.S.A.); aalmushaigeh@kacst.edu.sa (A.M.A.); 3Energy and Water Research Institute, King Abdulaziz City for Science and Technology, Riyadh 11442, Saudi Arabia; bmalotaibi@kacst.edu.sa; 4Chemistry Department, College of Science, Jouf University, Sakaka 2014, Saudi Arabia; amenad@ju.edu.sa; 5Electrical Engineering Department, Faculty of Engineering, Jouf University, Sakaka 2014, Saudi Arabia; agalharbi@ju.edu.sa

**Keywords:** carbon fibers, polymer-matrix composites (PMCs), thermoplastic resin, surface treatment

## Abstract

Carbon fiber-reinforced polymers are considered a promising composite for many industrial applications including in the automation, renewable energy, and aerospace industries. They exhibit exceptional properties such as a high strength-to-weight ratio and high wear resistance and stiffness, which give them an advantage over other conventional materials such as metals. Various polymers can be used as matrices such as thermosetting, thermoplastic, and elastomers polymers. This comprehensive review focuses on carbon fiber-reinforced thermoplastic polymers due to the advantages of thermoplastic compared to thermosetting and elastomer polymers. These advantages include recyclability, ease of processability, flexibility, and shorter production time. The related properties such as strength, modulus, thermal conductivity, and stability, as well as electrical conductivity, are discussed in depth. Additionally, the modification techniques of the surface of carbon fiber, including the chemical and physical methods, are thoroughly explored. Overall, this review represents and summarizes the future prospective and research developments carried out on carbon fiber-reinforced thermoplastic polymers.

## 1. Introduction

In an ever-evolving world, developing new sustainable materials with excellent properties while ensuring they fall into the category of circular economy materials is essential to meet industrial demands and prevent environmental pollution. New materials must overcome existing challenges such as high cost, recyclability, reliability, and energy consumption. For example, such materials for high-performance products need to be lightweight and strong to take diverse loading conditions, such as turbine blades in wind energy applications. They also must not create new problems regarding safety, availability, and processability. One of the main challenges of developing a new product is reducing the weight and increasing load-bearing capability at the same time [1,2,3,4]. One of the promising lightweight materials is carbon fiber (CF), characterized by high-strength, high-temperature resistance, and good chemical resistance. CF is non-toxic, low-density, has high wear resistance, and is a non-corrosive, recyclable material with an outstanding strength-to-weight ratio. Overall, it has exceptional thermal, mechanical, and electrical properties. CF is made when source materials such as synthetic polymers (polyacrylonitrile, pitch resin, or rayan spun) are carbonized through oxidation and thermal treatments (hydrolysis) at high temperatures while applying tension with final CF products’ appropriate controlled properties. It is well known that higher carbonization temperatures (up to 2500 °C) can achieve a high carbon content in CF. Today, CF-reinforced polymer matrix composite products are widely used in various applications due to their excellent mechanical, thermal, electrical, structural, and tribological properties. These applications include use in wind energy, aerospace, automobile, infrastructure, marine, and building and construction industries, as well as in sporting goods [4,5,6,7].

The global CF-reinforced polymer matrix composites demand is shown in Figure 1. This figure shows significant CF use in several industrial applications. CF can be classified into several categories depending on the properties, precursor materials, and final heat treatment temperatures. They can also be classified based on length or orientation within a matrix as long, short, continuous, and discontinuous fibers. Consequently, the various architectures of CF enable different applications. For example, discontinuous fiber composites are used in high-volume applications where nearly isotropic mechanical properties are desirable. Continuous fiber (CCF) composites are best used in low-volume applications that require maximum mechanical properties in one or two directions, such as impact panels, support beams, and containment vessels. Several studies reported the effect of CF’s orientation on the properties of polymer composites [8,9,10,11,12,13,14,15,16,17,18].

Carbon fiber reinforced polymers (CFRP) have been widely investigated. Many types of research have focused on using CF-reinforced thermosetting polymers such as epoxy and polyester resins. Many published reviews have explored state-of-the-art CF-reinforced thermosetting polymers. Moreover, manufactured thermoset composites are unrecyclable due to thermosetting polymers’ characteristics. Hence, in large-scale production aspects, they exemplify environmental and economic issues [19]. A recent review by Hegde et al. [17], who reviewed CFRP materials and their mechanical performance, stated that such materials’ prices considerably dropped in the 1990s. Subsequently, these materials were utilized in sports equipment. Additionally, between 1998 and 2006, the utilization of CFRP doubled in the world market. The compound annual growth rate for the utilization of CFRP in 2018 was predicted to be 12.5%.

On the other hand, another class of promising lightweight materials is thermoplastic polymers. They are also called thermosoftening plastics that become modulable at certain temperatures and solid upon cooling. Most thermoplastic polymers are recyclable and easily shaped to the desired requirements. Thermoplastic polymers can be combined with unidirectional CF, discontinuous (short and long CF), or CCF to achieve composite materials with improved mechanical, thermal, and electrical properties in one or multiple directions. Thermoplastic polymers can further be classified as the following: (1) commodity or general plastics such as polyethylene (PE), polypropylene (PP), polystyrene (PS), and acrylonitrile butadiene styrene (ABS) resin. (2) High performance or engineering plastics including polyamide (PA), polyethylene terephthalate (PET), polycarbonate (PC), polyetheretherketone (PEEK), polyetherimide (PEI), polyethersulfone (PES), and polyphenylene sulfide (PPS) [1,4,7]. Table 1 shows some thermoplastic polymers, chemical formulas, and related applications. Therefore, thermoplastic polymers have received vital consideration as a matrix due to the lack of prerequisites in curing stages and less dangerous chemical compositions, and better recycling suitability and mass production capability compared to thermosetting polymers. These characteristics give thermoplastic polymers an advantage over thermosetting polymers. Furthermore, the final composite products have enhanced properties compared to the individual components, i.e., thermoplastic and CF. Carbon fiber-reinforced thermoplastic polymers (CFRTP) offer weight reductions of about 50% compared to steel and 20% compared to aluminum [5,20,21].

CFRTPs are frequently manufactured using conventional molding approaches, such as injection, rotational, extrusion, vacuum, and compression moldings. Although CFRTP has attracted many researchers recently due to its excellent mechanical and thermal properties, recyclability, flexibility, less production time, and environment-friendly manufacturing, it is still in the development stages for some applications, and there are existing issues with high manufacturing costs to be overcome [1,7,19].

It is well known that synthesized CF materials have a smooth, natural surface with chemical inertness and are non-polar, while the polymer is generally polar. Due to this different polarity, the reinforcing process must be preceded by treating the CF’s surface. The treatment is conducted by creating functional groups on the surface of CF to ensure good interfacial adhesion between the polymer (matrix) and the CF (reinforcement), which is required to achieve high-performance composite materials; this is essential to their practical application. Many researchers have noticed the importance of strong bonding between the reinforcement and the matrix for high-performance composites [21,22,23,24,25].

Moreover, during the manufacturing process, many aspects must be taken into account to ensure a high quality of the final product while maintaining an efficient manufacturing process. For instance, the manufactured CF must be wear resistant, handle loads without cracking, and function successfully in a wide range of conditions such as high temperatures and humidity. Additionally, during the manufacturing process, energy consumption, cost of equipment and labor, environmental sustainability, and large-scale production ease are essential factors that must also be taken into account [4,25,26,27]. To improve the composites’ potential in the mentioned sectors and others, it is important to make a strategic road-mapping activity. Europe has a competitive composites industry. However, many challenges are still to be addressed. A roadmap for the challenges and the industrial uptake of CF and advanced high-performance composites’ supply chain in Europe has been published recently by Koumoulos et al. [25].

In this review, we explore the state-of-the-art of CF-reinforced thermoplastic polymers and their future, focusing on the modification methods for CF reinforcement. These possible modification processes are needed to improve the interfacial adhesion between the matrix and the CF. In conclusion, this review offers a comprehensive overview of the CFRTP properties necessary for scientists and decision makers to decide if CFRTP is a suitable material for their objectives and how this composite material could be utilized to achieve a more sustainable and circular economy of the materials for several high-performance applications including in the automotive aerospace industries and in turbine blades used in wind energy.

## 2. Properties of CFRTP

Researchers and developers have shown a great deal of interest in CFRTP composite due to its tremendous and wide range of properties and the potential of utilizing it in many industrial applications. Moreover, these properties can be altered or enhanced by determining which materials and methods to use. For example, what is the length of the fibers? In which direction are they aligned in the matrix? Was the surface of CF treated or not? Every choice made during the process will affect the composite properties; hence, it will either limit or expand the possibility of utilizing the material in specific industries. Some of these properties are crucial in every thinkable application, such as the mechanical strength of CFRTP.

Meanwhile, enhancing the electrical conductivity needed in specific industrial sectors such as electronics, energy storage, or in the automotive industry, when it is being used as a multifunctional part in addition to electromagnetic shielding effectiveness (EMI shielding effectiveness), is crucial when the composite is meant to be used in an application that requires an electromagnetic attenuation material, for example, when it is used in the wings of airplanes to protect them from lightning strikes. The following sections will explore CF surface modification and CFRTP’s properties as reported in the previous literature. It is essential to keep in mind that certain properties were enhanced for specific applications. Therefore, what seems like an excellent result for a particular application might be viewed as an obstacle in another sector, as shown in Figure 2.

### 2.1. CF Surface Modification: A Primary Factor Affecting the Performance of CFRTP

The interfacial property is a primary factor because when the bond is strong, the load is transferred successfully from the matrix to the CF without causing any damages to the product. The interfacial bond between the CF and the thermoplastic matrix is seemingly weak due to their unidentical polarities. Thermoplastics are mostly polar, while CF is not. Several CF surface treatment methods have been investigated to solve this issue, including both chemicals and physical treatment approaches [7,12,21,22,24,28,29,30,31,32,33,34,35,36,37]. It has been reported that the adsorption of some polymeric particles using the electrophoresis process could be used for controlling the interfacial properties and adhesion between carbon fibers and thermoplastic resins through the control surface adhesion between CF and polymer matrix [38]. Figure 3 displays a schematic diagram showing bad and excellent interfacial adhesion between the polymer matrix and CF. Figure 4 shows the most common treatment methods of CF surfaces used in this field.

#### 2.1.1. Chemical Treatments

**Coupling agents** and compatibilizers improve the adhesion characteristics in bonds between the reinforcement and matrix in composite materials. For instance, the addition of three types of maleic anhydride grafted polypropylene (MAPP) as coupling agents with different molecular weights and maleic anhydride contents were investigated by Wong et al. [28]. They studied their effects on the interfacial adhesion of recycled carbon fiber (RCF)-reinforced PP composites. They concluded that the compatibility was strongly dependent on the molecular weight and anhydride groups in the coupling agent. Various coupling agents also have been investigated by Han et al. [39], who used a silane coupling agent for CF-reinforced PP composites. The authors concluded that treating the CF surface using coupling agents significantly impacts the CF-PP matrix interaction and composite materials’ performance. A similar study has been carried out by Unterweger and his colleagues [35], who evaluated the influence of short carbon fiber (SCF) surface properties and the amount of modified homo PP (MAPP) as a coupling agent. Strong adhesion between CF and PP matrix has been reported by Cho et al. [40], who used a bi-functional group grafted as a coupling agent to modify long carbon fiber (LCF) to enhance PP/LCF composites’ mechanical strength. An aminated polyphenylene sulfide (PPS-NH2) as a compatibilizer agent has been investigated by Zhang et al. [41], who studied PPS/CF composites. The results showed that such amination chemical treatments improved the compatibility between the CF and PPS matrix, which enhanced the adhesion at their interface. Additionally, three different PE copolymers as compatibilizers have been used by Savas et al. [42], who fabricated a CF-reinforced high-density polyethylene (HDPE) matrix. It was found that interfacial adhesion depended on the type of copolymers. However, all compatibilizers’ addition improved interfacial properties of the investigated composites compared with the composites without any compatibilizers. Park et al. [43] investigated PC/CF composites’ interfacial properties using two coupling agents, including tetrahydrofuran-soluble graft copolymers and a water-dispersible coupling agent. His results indicated that both copolymers caused an enhancement in the interfacial shear strength (ILSS) due to chemical bonding at the interface between the functional group at the surface of CF and the groups in the copolymers. Liu et al. [44] stated that an enhancement of surface energy between CF and PVDF had been achieved after using a novel maleic anhydride grafted PVDF as a coupling agent. In a similar study, three different modification techniques were combined by Tran et al. [45], who fabricated PVDF/CF composites. The PVDF was modified by a maleic anhydride grafted PVDF, and the surface of CF was treated by electrochemical oxidation and/or epoxy sizing materials. The sizing materials are a coupling agent coating coated over a CF surface to improve CF’s binding capacity to the polymers. The term sizing is frequently used to remove the confusion between the coupling agent’s size and the size relating dimension [36].

Consequently, sizing materials have been applied to modify the surface of CF. The properties of CF-reinforced polyamide 6 (PA6) Composites have been evaluated by Karsli et al. [46], who used different sizing materials including polyurethane (PU), PA, polyimide (PI), phenoxy, and epoxy/phenoxy. The results obtained in this study confirmed that the selection of sizing materials had a critical effect on the final properties of CF-reinforced polyamide 6,6 (PA6,6) composites. As a result, PA and PU were determined to be suitable CF sizing materials for the PA6,6 matrix compared with phenoxy and epoxy/phenoxy. In an attempt to produce high-quality PA6 incorporated with long CF (LCF) composites, Luo et al. [47] used an isocyanate modified epoxy emulsion and silane coupling agent as sizing treatment of CF first to achieve the desirable interfacial bonding between CF and PA6. Another set of sizing material, i.e., epoxy/phenoxy, PI, and phenoxy and their effect on the mechanical properties of CF-reinforced PC composites, has been investigated by Ozkan et al. [48], who concluded that these sizing materials protected CF during the processing leading to better interactions between sized CF and PC matrix. In another study, PC polymer was used as a sizing agent for PA6/CF composites by Zhang et al. [49]. It was concluded that the sizing alters the chemical composition of the CF surface, including the (oxygen/carbon) O/C ratio, and the percentage of activated carbon atoms gradually increases as the sizing concentration increases. The interfacial strength between CF and PA6 matrix improved remarkably. Considering the composites’ interfacial strength, the most effective sizing concentration is determined to be 1.0–1.2%. The transverse fiber bundle test was completed to determine the interfacial adhesion between CF and PA6 matrix. Liu et al. [50] prepared poly (phthalazinone ether ketone) (PPEK)/CF composites, and they used PPEK as a sizing material for CF with three different concentrations: 0.1, 0.5, and 1 wt%. They studied the compatibility between sized CF and PPEK resin using contact angle and found that the uniformly sized CF was more compatible with PPEK resin than unsized CF. A similar result was observed by Wen Bo et al. [51], who investigated the interfacial properties of CF-reinforced PPEK composites. In contrast, Yu et al. [52] reported weak interfacial adhesion between the PC matrix and CF reinforcement after coating its surface by PET following by coupling agent treatment. A similar observation was noted for CF-reinforced PP composites by Unterweger and his colleagues [35]. It has been reported that, due to the complexity of the CF’s surface, sizing material composition, and differences in chemistry, a more comprehensive investigation into a cross-section of sizing materials should be conducted in future research [53].

Acid treatments have also been utilized to improve the interfacial properties of CFRTP composites. For example, the chemical interaction of HDPE with CF without using any coupling agent has been reported by Khan et al. [54]. They concluded that significantly proliferated fracture strain, flexural modulus, and flexural strength increased the CF layers in composites up to 20 layers. However, Shengbo et al. [55] reported that treated CF (with nitric acid) is more compatible with the HDPE matrix than the untreated one. They explained that the hydrophilic carboxylic group of benzoic acid reacted with a hydroxyl group of treated CF, which improved their compatibility with the PE matrix. Additionally, Chunzheng et al. [56] reported an enhancement of the interfacial interaction between ultra-high-molecular-weight polyethylene (UHMWPE) and CF after acid treatments. The CF was exposed to nitric acid oxidation treatments and introduced into polyoxymethylene composites (POM/CF) in an experiment conducted by Zhang et al. [57]. They concluded that the introduction of reactive functional groups, the surface’s roughness, and increased disordered carbon on the surface of nitric acid-treated fiber were proved. The nitric acid treatment altered the fiber’s surface roughness in a way that significantly enhanced the flexural strength and modulus relative to virgin POM for POM/CF composites. Liang et al. [58] used chloroform as a solvent for treating the surface of SCF.

Two methods of improving the interfacial interaction between CF and PS were investigated by Li et al. [59]. They used the modification of the PS matrix by adding the compatibilizing agent maleic anhydride grafted PS and functionalization of the CF surface with nitric acid. As expected with the surface treatment of carbon fiber, the authors concluded that the surface oxygen and nitrogen content increased, leading to a rise in the overall surface energy of the fibers, which resulted in an excellent interfacial interaction between CF and PS matrix. Both physical and chemical techniques (oxidation and coating) were used by Yan et al. [60], who modified the surface of CF with the oxidation method and coated it with a layer of PA12. The results showed improvements in dispersion and interfacial bonding between CF and thermoplastic matrix after these treatments. Qiujun et al. [61] manufactured UHMWPE/CF composite materials; they used acid-treated carbon nanotubes (CNTs) to enhance the resin, and CF’s interfacial adhesion is a combination of physical and chemical treatment methods. CF was immersed in CNTs solution. The results showed a significant improvement in interfacial interaction. This enhancement was attributed to CNTs interlock, which improved the compatibility between the UHMWPE matrix and CF reinforcement.

The electrochemical method was used to modify the CF’s surface by Shengbo et al. [55], in which packs of CF were connected to positive electrodes and immersed in nitric acid (HNO_3_). Their results indicated an enhancement in the interfacial strength due to CF’s functionalized surface. Li et al. [62] found that HNO_3_ treatment efficiently improved the CF-reinforced ABS composites’ interfacial adhesion. Additionally, ozone treatment was found to increase the oxygen concentration on the CF surface, which improved the interfacial adhesion with the matrix, by Fu et al. [63]. They reported no changes in the other properties, such as the tensile strength of the fibers themselves. Ozone modification and air-oxidation modification were used to improve the interfacial adhesion of CF-reinforced PI composites by Li et al. [64]. They found that ozone treatment effectively improved the interfacial adhesion between CF and PI. The strong interfacial adhesion of the composite made CF not easily detachable from the PI matrix and prevented the rubbing-off of PI, which, accordingly, improved the friction and wear properties of the composite. Similar observations were reported by Li et al. [65] for PA6/CF composites.

#### 2.1.2. Physical Treatments

Furthermore, **plasma treatment** for treating CF surfaces was reported by Montes et al. [66], who studied its effect on the interfacial properties of CF-reinforced PC composites. The authors concluded that the interfacial adhesion between CF and PC increased due to increased functional groups after the plasma treatment. A plasma treatment (oxygen and helium) at atmospheric pressure for different time treatments was conducted by Xie et al. [67]. The researchers found an improvement in the PA6/CF composite systems’ interfacial properties due to the increase in oxygen concertation in the surface of CF, roughening, and its energy surface. Additionally, microwave plasma treatments on CF have been reported by Lee et al. [34]; they reported an improvement in the interfacial properties of CFRTP composites due to the enhancement in the mechanical interlocking between the modified CF and cyclic butylene terephthalate (CBT) matrix. Lee et al. [68] investigated the effects of RCF plasma treatment at dry air and CO_2_ plasma conditions. They reported that the gas types and exposure time are the main factors for the modification’s efficiency for improving the adhesion properties. Recently, the effect of the atmospheric plasma treatment of RCF-reinforced PP composites at different plasma powers was investigated by Altay et al. [69]. Similarly, Han et al. [39] concluded that the hydroxyl group’s density was the highest among the specimens with 1 min of plasma treatment, but it reduced the plasma treatment time (at 3 min). Moreover, it has been reported that low-pressure plasma treatment was found to increase the amount of oxygen on the CF’s surface [67,69].

Additionally, the irradiation method is to enhance the adhesion performance of CFRTP composites. For example, an irradiated PP was used by Karsli et al. [70] as a compatibilizer for fabricating PP/CF composites. Their results concluded that the irradiated PP as a compatibilizer enhanced the interfacial adhesion between the CF and PP matrix, leading to improved mechanical properties. A similar achievement for adhesion has been reported by Mao et al. [71]. They used amino-functionalized CF using an electron beam irradiation technique, which gave a better surface and interfacial properties of their composites. An overview of new oxidation methods for PAN-based CF conducted by Shin et al. [24] selected PAN precursor fibers as the subject of focus and studied three major categories of radiation-induced polymer stabilization processes: electron beam, γ-radiation, ultra-violet, and plasma treatment. Therefore, it is concluded that further development of these radiation-based oxidation processes can significantly improve the speed of CF production and reduce its environmental impact. Additional recent studies by Jung et al. investigated the radiation effect on CF-reinforced HDPE composites [72], who concluded that the adhesion between CF and HDPE was improved, and the surface properties of CF HDPE were changed by irradiation. It was also supposed that irradiation provided two main effects on CFRTP. One was cross-linking of thermoplastic resin for efficient load transfer from resin to CF and the formation of surface functional groups and attractive interaction of these functional groups at the fiber and matrix interface.

**Materials coating** is also one of the methods that have been used to improve interfacial adhesion. For instance, metal coating with Ni-plated CF improves the desirable properties and the interfacial adhesion reported by Lu et al. [73], who manufactured different commodity plastics and reinforced CF composites, respectively. Still, this is not a favorable choice due to increased possibilities of corrosion occurrence. This degradative process is attributed to the electrically conductive nature of CF and its surface chemistry. The Ni-plated CF was used as a tracer to investigate the CF orientation in thermoplastic/CF composites by Nagura et al. [10]. Ofoegbu et al. [33] carried out a recent review that highlighted the potential corrosion challenges in multi-material combinations containing CFRP, the surface chemistry of carbon, its plausible effects on the electrochemical activity of carbon, and, consequently, the degradation processes on CFRP.

Furthermore, there are examples of grafting nanoparticles to the surface of CF to enhance the interfacial adhesion between CF and thermoplastic matrices. For instance, excellent interfacial adhesion has been reported by Li et al. [74], who investigated the performance of PES/SCF composites. They coated CF’s surface with graphene oxide (GO) and reported a remarkable improvement in the interfacial adhesion properties. Similar results were observed by Wang et al. [75], who investigated the effect of GO coating on PP/SCF composites’ properties. These improvements were attributed to the excellent progress in the chemical and mechanical interaction (interlocking) between GO on the surface of CF and the PP matrix as superior interfacial bonding. Yongqiang et al. [76] deposited CNTs and GO on CF. They reported an enhancement in the interfacial adhesion, roughness, and wettability of the CF surface to increase adhesion between the PI matrix and the CF due to hydrogen bonding and mechanical interlocking.

Ma et al. [77] prepared PA6/CF, and to enhance compatibility between CF and PA6, they modified the surface of CF by grafting GO on its surface using a coupling agent. The results showed an enhancement in the interfacial properties compared to untreated specimens. Li et al. [78] coated SCF with GO to enhance the PES/CF composite’s interfacial properties. They found an enhancement in the composite’s interfacial and mechanical properties compared to uncoated SCF with 0.5 wt% as the highest concentration of GO to enhance the composite properties. Irisawa et al. [79] studied the effect of nanofiber for CF-reinforced PA6 and concluded that nanofiber’s addition increased the bending properties of such composites. These results showed that nanomaterials are a promising candidate for improving interfacial adhesion between thermoplastic and CF to achieve maximum mechanical properties. A better interfacial adhesion for CF-reinforced vinyl ester composites was observed by Li et al. [80]. They concluded that better interfacial adhesion between CF and PEEK matrix was noticed after adding/grafting CNTs to the CF-reinforced composites’ surface. An article that reviews interfacial bonding techniques used to increase the fiber-matrix interfacial bond strength of CF-reinforced polyaryl ether ketones (PAEK) polymer has been published by Veazey et al. [81].

### 2.2. Mechanical Strength and Modulus of CFRTP

Due to excellent mechanical properties, the use of CF has grown remarkably. The CF-reinforced thermoplastic composites enhanced mechanical properties of final composites, including tensile strength, tensile modulus, flexural modulus, flexural strength, creep resistance, wear resistance, and toughness alongside other properties such as thermal and electrical conductivity. In the automotive, aerospace, and many other manufacturing industries, the usage of CF-reinforced polymers has rapidly improved in the last ten years due to the features mentioned above. However, the CF-reinforced composites have low wettability with most polymers because of their nonpolar surface characteristics. The low-interfacial bonding strength between the fibers and polymer matrices results in inadequate mechanical performance in composites [17,35,56,63,69]. The apparent ILSS of the composite is usually used to characterize adhesion quality between the fiber and matrix [39,51,56]. Likewise, a transverse fiber bundle test technique has been proposed to assess the fiber/matrix interfacial adhesion without making composite materials [48,82].

The ILSS increased by 300% for PP/SCF composites prepared with the addition of CNTs and MAPP as a coupling agent, as reported by Arao et al. [83]. Additionally, an increase of 115.4% and a 27% increase in impact toughness have been reported after grafting CNTs on CF as hybrid fibers for a CF-reinforced PPEK composite by Liu et al. [84]. Wen Bo et al. [51] reported that about 80% of the apparent ILSS in the PPEK/CF composite system was attributed to residual radial compressive stress at the fiber/matrix interface. A similar conclusion has been reported by Qiujun et al. [61], who studied the mechanical properties of CF-reinforced UHMWPE composites and stated that a 70% increase in ILSS was observed. Ma et al. [77] found that the ILSS increased by 40.2% of PA6/CF composites when GO was grafted onto CF’s surface compared with unmodified composites. Li et al. [78] investigated the mechanical properties of PPS/SCF composites. The maximum improvement was 12.1% and 31.7% for the tensile strength and Young’s modulus, and the maximum gains were 12.4% and 17.3% for the flexural strength and flexural modulus, respectively. These improvements were attributed to existing of GO in the composites. Khan et al. [54] reported that flexural properties of HDPE increased significantly upon increasing the layer of CF in the composites. Wang et al. [75] found that all flexural, tensile, and impact strength of PP increased by about 43% upon addition of 10 wt%.

Han et al. [39] reported an increase of up to 47.8% in ILSS in treated composites using coupling agents followed by plasma treatments compared with the untreated composites. Other mechanical properties were also enhanced after the treatment. A study was carried out by Tran et al. [45] in which they concluded that the ILSS increased by 184% due to the enhancement in the interfacial bonding for CF-reinforced poly (vinylidene fluoride) (PVDF) composites. Liu et al. [50] reported that CF improved the ILSS after coating its surface with PPEK. The authors also stated that the value of ILSS for sized CF was about 51.50 MPa, higher than the unsized CF, which was around 39.50 MPa.

Moreover, the enhancement of surface energy and mechanical interlocking between CF and PVDF has been investigated by Liu et al. [44]. The authors noticed that the surface of CF roughness and H-bonding improved and wettability of the treated CF-reinforced PVDF matrix. Additionally, they found that the flexural strength and modulus of the composites containing modified CF were also improved by 47% and 74%, respectively, compared to unmodified CF did. Connor et al. [85] found an increase in ILSS of 33% for the CF-reinforced nylon composites. Park et al. [43] reported an improvement of ILSS up to 70% for PC/CF composites. These variations in ILSS values could be attributed to several factors including interfacial area properties, matrix type, CF type, CF loading, and surface treatments.

Li et al. [86] studied the tensile properties of a treated CF-reinforced ABS matrix and reported that when the oxidized SCF content increased in the ABS matrix from 10 to 30 wt%, the tensile strength and tensile modulus improved significantly. However, these properties enhanced dramatically when PA6 was blended with ABS. Similar observations were reported by Li et al. [62] for an ABS/AP6 blend composites reinforced SCF system. A similar observation was noted by Anish et al. [87], who found that CF enhanced ABS’s hardness and compression strength in different wt%. However, ABS/CF in 30 wt% exhibited the best mechanical results. These improvements were due to better enhancements in the surface properties of CF reinforcement.

A different blend of polymers was fabricated by Zhou et al. [88], who studied the effect of CF reinforcement (5, 10, and 15 wt%) on the mechanical properties of PA6/PPS blend composites. This study’s mechanical results showed that the strength, modulus, wear resistance, and hardness of the composites improved significantly, although the strain values at break and impact strength were slightly decreased. A similar study was carried out by Luo et al. [89], who studied the effect of SCF on composites based on a PPS/polytetrafluoroethylene (PTFE) blend as a polymer matrix. They concluded that the strength, modulus, hardness and wear resistance, the elongation at break, and hardness of the PPS/PTFE composites were improved by introducing 15 vol.% of CF compared with the unreinforced blend matrix. Different composites based on polymer blends have been carried out by Zheng et al. [90], who prepared a blend of PEI/PES thermoplastics and introduced them into poly(phthalazinone ether sulfone ketone) (PPESK) polymers as a matrix and CCF as the reinforcement of PPESK/CF composites. They investigated both PPESK/PEI/CF and PPESK/PES/CF composites’ mechanical properties. The results indicated that the mechanical properties were enhanced. The improvement was attributed to good interfacial adhesion and low porosity resulting from PEI and PES’s addition into the PPESK matrix. This also improved the rheological properties of the PPESK matrix and gave enough impregnation during the preparation process of the composite materials.

Sharma et al. [15] studied the effect of the CCF orientation on PEI/CF composites’ mechanical properties with a loading of 80 vol.% CF, and the orientation angles were (0°, 30°, 45°, 60°, and 90°). Their results indicated that CF orientation influenced mechanical properties remarkably, including Young’s modulus, toughness, Poisson’s ratio, and percentage strain with respect to the loading direction. For instance, composites having CF in a direction parallel to loading (0°) proved most beneficial, while fibers beyond 45° deteriorated in performance excessively. Fibers at 75° were shown to have the poorest strength properties, followed by those at 90°. In conclusion, the aligning of fibers in a proper direction leads to better mechanical properties.

Luo et al. [47] prepared LCF-reinforced PA6 composites. The authors concluded that the tensile strength of PA6 composites containing LCF was much higher than the tensile strength of other composites having SCF by 24%. These enhancements were attributed to the excellent adhesion properties between PA6 and CF and the high aspect ratio of such LCF. The optimal loading of CF and sizing was 20 and 22 wt%, respectively, which exhibited the highest flexural and tensile strengths of PA6/LCF composites.

The tensile properties of poly(trimethylene terephthalate) (PTT)/CF composites have also been investigated by Vivekanandhan and colleagues [91], who reported an enhancement in mechanical properties as the concentration of CF increased. The addition of 30 wt% of CF into PTT resulted in significant tensile enhancement up to 120% and flexural strength up to 30% compared to neat PTT.

Yan et al. [60] fabricated PA12/CF composites to investigate their mechanical properties upon the addition of CF loading. They concluded that the incorporation of 50 wt% CF greatly enhanced both flexural strength and modulus of the investigated PA12/CF composites by 114% and 243.4%, respectively. Yan et al. [92] reported the flexural properties of PA6/CF composites. They stated that the addition of 30 wt% CF significantly enhanced flexural strength and modulus by 208% and 438%, respectively. These improvements in the flexural properties were attributed to good interfacial bonding, filler dispersion, and surface CF chemistry after surface treatment of CF.

Ma et al. [93] prepared a higher performance unidirectional CF-reinforced PA6 thermoplastic composite and investigated its mechanical properties. The results indicated that excellent tensile properties, including tensile modulus and strength of PA6/CF composites and uniform CF distribution, have been proved. Similar results have been reported for CF-reinforced nylon composites by Hassan and colleagues [94] and Dickson et al. [16].

A comparison between the mechanical behavior of PA6/CF and PA66/CF composites was carried out by Botelho et al. [95]. They concluded that both matrices showed slight mechanical behavior improvements, including tensile, compressive, and ILSS behavior when reinforced by both fabric and unidirectional CF. The microscopic damage progress in both composites was observed through optical and scanning electron microscope techniques. It was shown that shear failure at the/PA/CF interface was mostly responsible for damage development, initiated at relatively low stress.

The performance of plastic gear made of CF-reinforced PA12 was investigated and compared with PA6/CF, PA66/CF, and PA46/CF composites by Kurokawa and colleagues [96]. The authors reported that the PA12/CF composites having grease showed an excellent load-bearing characteristic among all investigated PA composites, and this load increased by increasing the molecular weight of PA12.

Wu and coworkers [97] reported that the interlaminar fracture toughness and transverse properties enhanced the PES matrix’s molecular weight. Their results also concluded that the CF distribution was uniform and with reasonably good wetting with the investigated matrix. This resulted in a higher longitudinal flexural modulus and PES/CF composites’ strength compared with the pure PES matrix.

Numerous studies have focused on the relationship between fiber length and the mechanical performance of CFRP. For instance, Karsli et al. [98] studied the effect of both loading and size of CF-reinforced PA6 on the tensile properties of resultant composites. Their results showed that increasing the CF loading led to improvements in tensile strength, modulus, and hardness, but reduced strain at the break values of composites. Meanwhile, the investigated length ranges of CF (0 to 50 µm) had no effect on these mechanical properties except that strain at break was improved. Kim et al. [99] analyzed the impact of CF length, CF loading, and processing speed on PU/CF composites’ mechanical properties. They concluded that these parameters have a substantial effect on the mechanical properties of the composites. It was noticed that the CF length decreased from 163 to 148 μm after the extrusion process. This indicates that the manufacturing process impacts the final properties of the prepared composites. Li et al. [82] stated some details about interfacial parameters between such materials, which may be useful for further simulation studies for optimizing all related parameters that affect mechanical properties such as CF length, loading and processing conditions, annealing temperature, and cooling rate. They used a transverse fiber bundle test which was proposed to assess the fiber/matrix interfacial adhesion without manufacturing composite materials. Furthermore, fiber length distributions have been reported to depend on the processing conditions by Fu et al. [100]. They investigated the fracture resistance of PP/CF composites under Charpy impact load. They concluded that the composite impact resistance was shown to depend on the CF’s length and hence on processing. The notched Charpy impact energy of the PP/CF composites increased with decreased CF loading in the PP matrix. Similar observations have been reported by Rezaei et al. [101], who studied the effect of fiber length on the thermomechanical properties of SCF-reinforced PP composites. Similarly, Unterweger et al. [102] evaluated the effect of fiber length distributions and content on SCF-reinforced PP polymer’s mechanical properties. They concluded that tensile strength, tensile modulus, and impact strength were improved upon the increasing amount of SCF despite reducing fiber length, tensile strength, tensile modulus, and impact strength. The longest CF in the final composites was reported in their study. This result seems to contrast with the result obtained by Kim et al. [99], who noticed that the tensile strength of the PC/CF composites decreased. This reduction was attributed to the decrease in the CF length after the fabrication process.

Ozkan et al. [48] considered the effect of SCF on tensile strength and modulus of the PU matrix. They found that the unseized CF improved the tensile strength and modulus of composites by 105% and 450%, respectively. In comparison, sized CF improved the tensile strength and modulus of composites by 150% and 540%, respectively, compared to the simple PU matrix.

Different thermoplastic composites were synthesized by Hwang [103], who investigated the effect of CF loading on the tensile strength of foamed and solid polybutylene terephthalate (PBT) composites. The author stated that the addition of 8 wt% loadings of CF to PBT resulted in improved tensile strength; however, further addition of CF led to a reduction in such property. This reduction is attributed to agglomeration/aggregation of CF into the PBT matrix and a reduction in the foams’ cell size.

Gabrion et al. [104], who fabricated CF-reinforced PI composites, investigated the influence of temperature on the tensile properties. The authors concluded that the tensile strength was higher than 1200 MPa in the fiber direction at a temperature range varying from −50 to 250 °C but with low ILSS at high temperatures. The material also had outstanding fatigue strength under tension in this material direction.

Yu et al. [52] reported that the strength and modulus of shopped CF-reinforced PET composites continuously increased along with a clear ductile–brittle transition by increasing the amount of CF with different length and aspect ratios. They stated that the tensile strength and modulus of PET/CF composites increased with an increasing aspect ratio of CF (under the same loading of CF); this increase accompanied the decreased impact strength and elongation at break 20 wt% of the CF. Hamilton et al. [105] reported that 20 wt% of CF could increase thermoplastic composites’ wear resistance based on the PEEK matrix. Similar observations have been noted by Karsli and Aytac [106], who reported an enhancement of wear properties by incorporating CF into PEEK composites.

Kada et al. [107], who investigated the effect of tensile properties of SCF-reinforced PP composites, reported that a 30 wt% SCF content in the PP matrix improved the tensile strength and modulus of composites by 455% and 168%, respectively, when a MAPP was used as coupling agent during the preparation process. However, without MAPP, although the tensile modulus increased, the tensile strength decreased due to poor adhesion between inert hydrophobic CF and the hydrophobic PP matrix. The MAPP treatment had a direct influence on the mechanical properties of the composite. Do et al. [108] investigated the effect of PP on PA6/CF composites’ mechanical properties. Their results showed that the investigated mechanical properties, including ultimate tensile strength, elastic modulus, and elongation at break, were exceptional for the composites containing PP compared to those composites without PP as a coupling agent. The composite with 30% PP had the lowest ratio of tensile strength and elastic modulus reduction by 18% and 15%, respectively, compared with the composites with 0% PP loading.

Cho et al. [40] analyzed the tensile and flexural strengths of PP/LCF thermoplastic composites by increasing the adhesion between the PP and CF matrix. The mechanical strength of the resulting composite was significantly enhanced. These improvements can indicate better interfacial bonding between fiber and matrix as discussed earlier in Section 2.1.

Cho et al. [109] investigated the mechanical properties of polyketone (PK)/CF novel composites, and they found an enhancement in Young’s modulus of 520% and tensile strength by 189% at 30 wt% CF content. In contrast, a significant decrease in the elongation at break was observed in the PK/CF composites even at very low loading (5 wt%) of CF.

The tensile strength of the as-received and plasma-treated CF-reinforced CBT composites was enhanced by ~362.5% and 436.3%, respectively, compared with that of the pure CBT matrix as reported by Lee et al. [34]. They incorporated 70 vol.% of CF into CBT without any defects and pores in the final composites. However, Tobias et al. [110] reported a 60% increase in the strength of the CF-reinforced CBT matrix when the latter was chemically modified with a small number of chain extenders. They stated that the composites samples show poor ILSS, and cracks were observed.

Different PC/vapor-grown carbon fiber (VGCF) composites prepared by Choi et al. [111] were investigated for their mechanical properties before and after the CF rolling process. Their results indicated that the mechanical properties improved significantly due to CF’s orientation within the PC matrix. In contrast, Carneiro and colleagues [112] found that the tensile properties improved after adding VGCF into the PC matrix, but the impact resistance property was reduced significantly. They suggested that the rolling process could be used for further improvements in mechanical properties. Recently, Maqsood et al. [113] characterized both CCF and SCF’s influence on the tensile strength of CF-reinforced polylactic acid (PLA). They concluded that the tensile strength and flexural stress increased by 460% and 121%, respectively, for composites containing CCF compared to those having SCF.

Li et al. [114] reviewed the analysis of the properties of the CF and the tensile and dynamic mechanical properties of the UHMWPE hybrid composites (charcoal wood and CF in the matrix). Young’s modulus with tensile strength was significantly augmented with increasing loading of CF. They increased by 415% and 46% correspondingly. However, the elongation/strain at break decreased substantially by 95%. There is no doubt that whenever we increase CF concentration, it always increases the storage modulus. The storage modulus reached ~20 GPa for the samples containing 8 wt% CF compared to ~2 GPa for unreinforced UHMWPE/charcoal samples at room temperature.

Unterweger et al. [115] provided a good overview of the mechanical and physical properties, cost, and reinforcement effectiveness of synthetic and thermoplastic fibers and their characterization. Most materials are affected in some manner by environmental effects such as temperature and humidity. The properties and characteristics may change, and the material could be degraded. Research has explored this aspect, and it has shown several outcomes [93,116,117,118,119,120,121]. Table 2 summarizes the variety of composite materials, modification techniques, and the obtained mechanical properties of CFRTP over the last decade.

### 2.3. Electrical Conductivity and Electromagnetic Shielding Effectiveness of CFRTP

Besides the great mechanical properties of CF, it can be used for other tasks based on its multifunctional properties, including electrical conductivity and electromagnetic interference shielding. These properties of CF used as reinforcement in composite structures are the basis for several multifunctional applications. The significance of carbon is the extremely stable hexagonal plane grid and the planes’ delocalized electron cloud. The deformation and separation of the hexagonal carbon rings require high energy, which provides the CF’s strength at the macro level. The free electrons in the electron cloud make it an excellent electrical conductor. The electrical resistance of CFRTP depends mostly on the type of material used (precursors), the manufacturing conditions, the crystalline structure of polymer matrices, and treatments [2,3].

Lu et al. [73] studied the preparation of CF-filled ABS composites and investigated their electromagnetic interference (EMI) shielding effectiveness (SE) and electrical conductivity with and without metal coating (thickness of the layer was 0.2–0.5 μm). With the increase in the CF content, the composites’ resistivity with the nickel-coated CF decreased. The further decrease in the composites’ resistivity with the same nickel-coated CF was higher than that with the uncoated CF. They reported a resistivity of around 10–4 Ωcm, an order of magnitude less than that for the uncoated fibers. The composites’ EMI SE with 10 vol.% content of the CF coated with nickel was found to be 50 dB.

Similarly, Huang et al. [122] reported an enhancement in PC/ABS/nickel-coated CF composites’ EMI shielding. The best EMI shielding effectiveness was about 47 dB. The same group also reported some promising results in enhancing the EMI shielding of a Ni-coated ABS/CF composite. The best **EMI shielding effectiveness** was 44 dB [123]. A similar composite was prepared by Nishikawa et al. [124], who reported the electrical properties with different CF loadings. The composites’ electrical resistivity decreased with an increase in the CF-reinforced ABS plastic content, and the critical volume fraction (percolation threshold value) was found to be 0.11 vol.%. The EMI SE was not as expected due to the composites’ low conductivity in the out-of-plane direction.

Rahaman et al. [125] reported the EMI SE of SCF-reinforced ethylene-vinyl acetate (EVA) and acrylonitrile butadiene copolymer and their blend composites. They reported a marginal increase in EMI SE with the increase in electromagnetic radiation frequency, but a sharp increase was observed with an increase in the SCF contents of 20 phr. Das et al. [126] reported an SE of 34.1 dB at a similar fiber loading (30 phr) in NR- and EVA-based composites. The authors stated that the composites having a CF loading of ≥20 phr could be used for EMI shielding applications.

Zhang et al. [127] aimed to improve the blend composite’s electrical properties by using an electric conductive reinforcement, VGCF. They used HDPE and isotactic polypropylene (iPP) (50/50) as a matrix. They reported an enhancement in the electrical conductivity and lower percolation threshold of CFRTP when the CF loading was 1.25 parts per hundred parts resin (phr), compared with the neat polymers. To explain the results, SEM was used, and they attributed the improvements to the particular locations of CF; in other words, the dispersion of the filler within the matrix plays a role in enhancing the composite’s properties. A similar VGCF was utilized by Choi et al. [111], who reported the electrical conducting properties of PC composite sheets reinforced with VGCF reinforcement. The composites’ resistivity was found to be 10 and 0.5 Ωcm at a VGCF content of 10 and 25 wt%, respectively. The dispersion of the VGCF in the polymer matrix was found to be homogeneous, and the electrical conductivities of the composites increased. Simultaneously, percolation threshold values decreased with an increase in the loading of the VGCF, leading to better conduction networks.

An ultra-low percolation threshold value was observed by Zhao et al. [128], who studied the effect of CF with a large aspect ratio of carbon black on the conductive properties of the PP composites. The addition of 0.155 vol.% CF resulted in a significant decrease in the percolation threshold value. The reduction in the percolation threshold observed in the scanning electron microscope was ascribed to the increase in interparticle contacts, resulting in developing a shish–calabash-like conductive network. A similar morphology observation was reported by Cipriano et al. [129], who investigated the influence of carbon nanofibers on the electrical properties of PS composites. They found that the unfilled PS matrix’s electrical conductivity was around 10–8 S/cm, which increased to about 10^−2^ S/cm for the composites filled with 15 wt% carbon nanofibers. The authors also reported that annealing processes could improve the electrical conductivities of composites at high temperatures. Similarly, Thongruang et al. [130] studied the effect of graphite filler on the HDPE/CF composites system. They demonstrated that the addition of graphite to the composites increased the conductivity compared to the composites without graphite. The conductivity increased from ~0.1 (Ωcm) to 5 (Ωcm) for a composite having 10 wt% CF compared to HDPE matrix; however, it jumped to 18 (Ωcm) for a composite containing 10 wt% CF and 50 wt% graphite. Microscopic analysis of the composites showed that the CF depicted favored alignment according to their length compared to the composite film’s thickness.

Liang et al. [58] reported composites’ resistivity prepared by incorporating SCF in ABS resin matrix. The ABS composites with an SCF loading of up to 2 vol.% showed no improvement in the conductivity of the composites. However, above 2 vol.%, the composites’ resistivity showed a steep decline in resistivity from 10^13^ to 8.83 Ωcm and 10^14^ to 884 Ωcm for the 6 mm and 3 mm length of SCF, respectively, thereby providing good electrical conductivity to the resultant composites. The ABS composites rapidly changed from the inductor phase to conductor at a critical percolation threshold value between 1 and 2 vol.%. Another study was carried out by Tzeng et al. [131], who reported the EMI SE of on ABS/CF composites coated with nickel and copper metals. The electroless nickel-coated CF-reinforced ABS composites demonstrated higher electrical conductivity. Hence, the better EMI SE ability compared to the copper-coated CF was due to the excellent bonding between the nickel coating and CF surfaces. It has been reported that double-layer metals covering CF increased the EMI SE of composites effectively [132]. Ozkan et al. [48] reported the electrical properties of SCF-reinforced PC composites. These composites’ highest electrical conductivity containing 30 wt% CF was around 0.0035 S/cm compared to about 0.0005 S/cm for a pure PI matrix. Additionally, Hong et al. [133] reported an EMI SE of 30 wt% for CF-reinforced PP composites with the addition of 1 wt% of carbon nanotubes. A decrease in volume resistivity and an increase in the PP/CF composites’ EMI SE were observed. Increasing the CF length from 200 to 250 μm in the PP composites showed the best results; moreover, a long blending time and high speed can lead to good CF dispersion in principle, but there was an optimal saturation point in this composites system. A similar study was conducted by Unterweger et al. [102], who evaluated the impact of CF length and content on PP/CF composites’ electrical conductivity. They concluded that electrical conductivity showed a strong dependence on the fiber length and showed a linear correlation with the weight and average fiber length in the investigated range of 100–350 μm. When the CF content was raised from 5 to 10 vol.%, more than two orders of magnitude were in the electrical conductivity. However, a further growth to 15 vol.% CF only had a minor impact on the conductivity.

Xi et al. [134] studied the electrical properties of SCF-reinforced PEs including both UHMWPE and low-molecular-weight polyethylene composites. An excellent positive temperature coefficient was achieved. The conductivity increased with an increase in the heat treatment time due to the formation of better reconnection of SCF networks in the polymer matrix. A similar study was conducted by Shen et al. [135]. They investigated the combined effects of carbon black and CF on composites’ electrical properties based on PE or a PE/PP blend. The volume resistivity of the HDPE/carbon black/CF and HDPE/PP/carbon black/CF with 2 wt% CF decreased by around 3.0 and 11.2 orders of magnitude, respectively compared to that of the HDPE/carbon black and HDPE/PP/carbon black composites. The intensity of the positive temperature coefficient and the temperature coefficient of resistivity of the HDPE/carbon black/CF and HDPE/PP/carbon black/CF composites increased significantly with increasing CF loading.

The negative temperature coefficients were neglected because CF is not as easily agglomerated as other reinforcement such as carbon black and graphite. Another study was carried out by Ameli et al. [136], who investigated EMI SE and the electrical conductivity of a CF-reinforced PP composite containing carbon black in two forms, i.e., solid and foams. At 5 vol.% CF content, both composite samples’ conductivity decreased proportionally with frequency in the whole range. However, the electrical percolation threshold was 8.75 and 7 vol.% for solid and foams composites, respectively. The dielectric permittivity improved, and the through-plane electrical conductivity increased by up to six orders of magnitude, resulting in an increase of 65% in these foamed composites’ specific EMI SE. These results indicate that processing and matrix form could affect the electrical properties of the final composite materials. Hwang [103] reported the EMI SE property of solid and microcellular (foamed)-injected PBT/CF composites with various fiber contents and aspect ratios. He found that the microcellular composites showed better electrical conductivity for any particular CF content than those of the solid ones. The foaming process distorts the CF’s orientation, increasing the end-to-end fiber contacts, thereby increasing the electrical conductivity. The composites showed almost no EMI SE at 13 wt% CF content; however, at 30 wt%, it improved significantly to around 10 and 11.16 dB for solid and foamed composites, respectively. This result indicated the advantages of foamed composites over solid one to enhance the final composites’ electrical conductivity.

Saleem et al. [137] reported CF-reinforced composites of PEEK and PES as polymer matrices. They observed that the percolation threshold for the PES/CF and the PEEK/CF composites occurred at a CF loading of 10 wt% and 35 wt%, respectively. The higher percolation threshold for the PEEK was because the PEEK is a highly crystalline polymer compared to PES. Hence, the formation of the conducting pathways is not as easy in PES. At the percolation threshold, the measured electrical resistivity for both the composites was around 10^6^ Ωcm. The authors observed that the heat treatment of CF at a higher temperature improved the graphitic structure, resulting in the CF’s better electrical conductivity. This observation is in good agreement with previous results reported by Xi et al. [134]. It has been reported that the electrical resistivity of the CF treated around 2000 °C showed a resistivity of around five orders of magnitude lower compared to the untreated ones [138]. Table 3 summarizes the variety of composite materials, modification techniques, and the obtained electrical properties of CFRTP over the last decade.

### 2.4. Thermal Stability and Thermal Conductivity of CFRTP

The primary thermal properties of CFRTP are thermal stability, thermal conductivity, melting temperature (T_m_), and glass transition temperature (T_g_). Researchers have investigated these properties extensively in an attempt to enhance them. The T_g_ of polymer composites normally depends on several factors such as the chemical structure and conformation of the polymers, degree of crystallinity, fiber dispersion, and interactions between the fiber and the polymer. Several studies have confirmed that the addition of fillers affects the T_g_ and the breadth of the transition due to changes in the mobility of the polymeric chains in the host matrix. By improving thermal properties, CFRTP becomes more suitable for fulfilling the already existing demands in various high-temperature sectors such as the aerospace, oil, and gas industries.

Kada et al. [107] reported the thermal properties of PP composites reinforced with varying quantities of SCF (9, 15, 20, 25, and 30 wt%). The results showed that the PP matrix and their CF composites exhibit a single-step decomposition. The PP molecular degradation started at around 408 °C, and the decomposition maxima occurred at 468 °C, and the maximum rate of degradation was 2.3%/min. An improvement in the initial degradation temperature was observed on the incorporation of the SCF into the PP matrix. Compared to the neat PP, the composites with 9% and 30% SCF content showed an increase of 10 °C and 20 °C in initial degradation temperature, respectively, which was attributed to the higher heat absorption capacity of the CF and the delayed decomposition temperature results from the reduced heat release rate of the CF. The enhancement in the composites’ thermal conductivity with an increasing volume fraction of the CF was due to the CF’s higher thermal conductivity than that of the PP matrix.

On the other hand, Yilmaz et al. [139] reported that the melting behavior of CF-reinforced PP composite was considerably influenced by the thermal history rather than the CF’s presence. Their results show melting over a wide range of temperatures, with two peaks appearing for the samples with no thermal treatment and those annealed at lower temperatures irrespective of the CF’s presence. Wang et al. [75] reported PP/SCF composite’s thermal properties containing 10 wt% of SCF. Their results exhibited that the composite exhibited improvement in thermal stability and crystallization temperature. Cho et al. [109] studied the thermal properties of the PK/CF composite and reported an enhancement of thermal conductivity up to 300% with an increase in CF content up to 30 wt% in the PK matrix. It has been reported that thermal stability increased in EVA, acrylonitrile butadiene copolymer, and their blend composites due to the restraint of their chain motion into the polymer composites generated by adding CF [125].

Khan et al. [54] reported the thermal properties of multi-layered laminated composite panels of CF-reinforced HDPE. The thermal degradation of the neat HDPE and the HDPE/CF composite showed a single continuous decline in the residual weight mainly due to the HDPE chains’ unsystematic scission. The composites’ thermal degradation temperature started at around 30 °C higher than that of the neat HDPE. Similarly, the maximum decomposition temperature for the composites was about 15 °C higher compared to pure HDPE. The multi-layered laminated composite panels’ thermal stability also improved by 41%, making these composites suitable for applications at higher temperatures.

In contrast, Thongruang et al. [130] found that SCF does not significantly affect the HDPE matrix’s thermal properties. They stated that the effect of long CF was more pronounced at high temperatures on the thermal properties. Additionally, Liu et al. [50] concluded that the CF coated by thermoplastic resin was more stable than untreated CF and had increased surface energy and wetting performance.

Rezaei et al. [101] used SCF of five different length sizes (10, 5, 2, 1, and 0.5 mm) as a reinforcing fiber (10 wt%) in PP as a matrix. They reported that compared to the shorter CF, the longer CF showed better thermo-mechanical properties as fillers in the matrix. The thermogravimetric analysis results showed that increasing the incorporated CF’s length led to an increase in the composites’ thermal stability. The glass transition temperature (T_g_) of composites combined with 10 mm length CF increased by 19.5% compared to that of the unfilled PP. Overall, the thermal degradation of the PP/SCF composites was improved for all investigated lengths compared to plain PP.

Gabrion et al. [104] studied two types of composite structures (plates and tubes) of a unidirectional CF-reinforced PI composite. The authors reported a longitudinal and transversal coefficient of thermal expansion of 1.7 × 10^−6^/°C and 30 × 10^−6^/°C, respectively, clearly depicting the materials’ high anisotropy. They reported that above 200 °C, the expansion with an increase in temperature was non-linear. Two transitions were observed at approximately 235 °C and 385 °C, attributed to T_g_ and T_m_ of the polymeric material, respectively. They reported that the weight loss depended strongly on the environmental conditions. The weight loss in the inert atmosphere due to degradation was significant above 500 °C; however, the degradation started at a lower temperature (~400 °C) in an oxidizing atmosphere. A similar study was carried out by Karsli et al. [52], who examined the performance of CF-reinforced PA6,6 composites. The initial decomposition temperature and the decomposition temperature at the maximum rate of the composite material were determined. It was found that the lowest decomposition temperature was about 270 °C, and the highest temperature at which no further weight loss was observed was approximately 500 °C for all composite samples used in their study.

Samyn et al. [140] reported the thermal properties of PI/CF composite materials. PI filled with 30 wt% CF showed an improvement in the heat distortion temperature or heat distortion temperature value by 10 °C. T_g_ and melting points showed no considerable change in the values with the incorporation of the CF. The thermal conductivity improved from 0.17 W/(mK) for the neat PI to 0.49 W/(mK) in the molding direction and 0.22 W/(mK) in the transverse direction for the composites. In a similar composite, Dong et al. [141] reported PI/CF composites’ thermal properties. The T_g_ value obtained showed an increase in the T_g_ values with the increase in CF content in the composites. At 5 vol.% CF contents, a slight decrease in T_g_ was reported but was found to increase to 241 °C and 244 °C at a CF content of 20 and 30 vol.%, respectively, compared to that of the neat PI value of 231 °C. The number of these confined chains increased with an increase in the CF content. The segmental motion and relaxations can occur only at higher temperatures and over a broad range of temperatures leading to an enhancement in T_g_ values. Similar observations have been reported by Vivekanandhan et al. [91], who fabricated PTT/CF composites. They reported that PTT composites containing 30 wt% exhibited an increase of more than 150 °C in the heat deflection temperature, and no significant changes in the melt temperature were observed. Additionally, Karsli et al. [98] investigated the effect of SCF content and its length on the thermal properties of CF-reinforced PA6 composites. The results showed no change in the values T_g_ and T_m_ for the composites even with the CF loading increase. However, the heat of fusion and the degree of crystallinity of the composites decreased with the rise in the composites’ CF loading. The higher fiber content restricts the mobility of polymer chains in the matrix and obstructs the crystal growth. On the other hand, CF length at the studied range did not significantly affect the thermal properties.

Yan et al. [92] studied PA6 composite with 30 wt% CF with a length of 7 mm and a diameter of 7 μm. The thermal conductivity of PA6 annealed at 80 °C and was 0.21 W/mK, and increased by about 24% when the annealing temperature was increased to 190 °C. The thermal conductivity of PA6/CF composites increased to 0.32 W/mK and improved by 13% in the annealing process. The heat deflection temperature value for PA6 lies between 64 °C to 77 °C based on its annealing temperature; however, for the PA6/CF composites, it was around 214 °C with a negligible effect for thermal annealing. The results also showed that the T_g_ of PA6 increased from 60 °C to 78 °C with the incorporation of the CF and thermal annealing, which led to an improved T_g_ for both the PA6 and the PA6/CF composites.

A variation in T_g_ reported by Munirathnamma et al. [142] characterized the PBT and PES polymer composites reinforced by CF. Their result showed a T_g_ of 44 °C for neat PBT, whereas the PBT composites containing 30 and 40 wt% CF showed a T_g_ of 46 and 44 °C, respectively. The incorporation of 30 wt% CF in the PBT matrix led to a nominal increase in T_g,_ suggesting a restriction on the matrix’s segmental chain mobility due to the CF’s presence. However, a T_g_ of 218 °C for pure PES decreased to 212 °C for composites containing 30 and 40 wt% CF. This reduction in T_g_ suggests compact molecular packing due to interface development. Connor et al. [85] reported the thermal properties of CCF incorporated in nylon. Filaments were used to fabricate composite layers by printing with a CF content of around 35 to 41 vol.%. The nylon filament composites showed no melting peak attributed to the addition of CF in the thermal processing history of this polymer.

Yu et al. [46] reported PC composites’ thermal conductivity with different loadings of chopped CF coated with PET and treated using 3-aminopropyl triethoxy silane. The results revealed that incorporating the chopped CF enhanced the thermal stability of composites by restricting the pyrolytic degradation of the polymer matrix. They reported an increase in the composites’ in-plane and through-thickness thermal conductivities with the increase in chopped CF content. The in-plane and through-thickness thermal conductivity for a 50% fiber content was 2.45 W/mK and 0.59 W/mK, compared to that of 0.20 W/mK for the neat matrix. Sun et al. [13] reported the thermal conductivity of a polysulfone/CF composite. The thermal conductivity was 1.82 W/mK at a CF loading of 26 vol.%. Similar results have been reported by Yoo et al. [143]. They stated a significant difference (up to 25 times) between the in-plane and the through-plane thermal conductivities of PA6 composites reinforced by CF. Saleem et al. [137] investigated the thermal conductivity of CF-reinforced PEEK and PES matrices. They stated that the thermal conductivity of the matrices improved upon the incorporation of 20 wt% CF. The further addition of CF resulted in a slight improvement in the conductivities. They also concluded that composites containing PES had better thermal conductivity due to the deficiency of crystallinity. Table 4 summarizes the variety of composite materials, modification techniques, and the obtained thermal properties of CFRTP over the last decade.

## 3. Future Prospects

In the future, CFRTP research and development activities will be focused on manufacturing techniques, recycling methods, cost reduction, and improving properties. These materials are attractive characteristics for various industrial applications as many types of thermoplastic polymers can be utilized as matrices for CFRTP composites. Ongoing developments in the processability and engaging CFRTP in terms of a cost-effective viewpoint and synergies between industrial sectors will pave the way to high-volume production that industries need to meet progressive demands. Advancements are also required for the preparation of efficient, cost-effective, and facile CFRTP materials. The progressive ideas are likewise vital for the high production rate of such materials. The overall growth of polymer composite materials having embedded functionality is anticipated to exceed five kilotons by 2029 [144]. CFRTP are lightweight polymer composite materials that show excellent properties and great potential for low-cost manufacturing when compared to thermosetting composite materials. As for CF, the worldwide industry manufacturing CF is set to grow enormously over the next ten years. This tremendous growth will be led by various factors, such as high demand for low-weight materials and creative design that needs solutions related to composite technology [145]. The future perspectives of CF are undoubtedly on a positive track. However, the integration of CF into huge markets and several common utilizations mainly depends on the capabilities of the manufacturers. CFRTP will achieve a wide potential if both polymer and CF manufacturers [146] continue to observe new applications and develop creative and low-cost technologies. The major CFRTP consumption sectors include the aerospace, defense, automotive, renewable energies, sports equipment, and construction industries [147]. For instance, aircraft makers have also been focusing on reducing the overall weight of aircraft so as to increase the efficiency of the product. The use of CFRTP helps in drastically lowering the overall weight. Rising demand for CFRTP as an alternative to metals such as steel and aluminum is expected to stimulate growth in such applications [148]. Similarly, the use of CFRTP-based parts is increasing compared with the use of metallic-based parts in airplanes and automobiles, which could decrease weight, greenhouse gas emissions, and consumption of energy. For example, CFRTP manufacturing processes have been recognized for automotive body panel applications including structural and non–structural components such as seat structures, bumpers, hoods, and fuel tanks [5,6,149]. In general, the strength and stiffness of a CFRTP material remain very much a function of the reinforcing material, but its mechanical natures are determined not only by the CF alone but by a synergetic influence between the CF and the polymer matrix. Particularly, the mechanical testing results of CFRTP materials can provide a 40–50 percent saving in weight for an equivalent bending stiffness in comparison with steel panels [150,151,152,153]. In terms of cost, CFRTP composite materials offer the cheapest processing technologies, as stated by Friedrich et al. [154] who suggested CFRTP composites as a future possibility in automotive applications. Another possible application for CFRTP is in pipelines in the oil and gas industries. The progress of reinforced thermoplastic polymer pipes for oil and gas applications have been reviewed by Morozov et al. [155], who stated that reinforced thermoplastic polymer pipes have been gradually recognized as a significant alternative to metallic pipes due to their diverse advantages such as a higher stiffness-to-weight ratio, enhanced fatigue resistance, and improved corrosion resistance. Venkatesan et al. [156] studied the mechanical properties of CF-reinforced composites in the deep sea. Their results showed that the investigated mechanical properties were not affected by the sea environment, and corrosion or degradation and bio-film formation was not observed. Therefore, CFRTPs are increasingly being utilized in oil and pipeline applications, especially those in deep water, as they can maintain their mechanical properties in seawater and provide additional cost savings in terms of strength-to-weight ratio in comparison with steel [157,158,159], which has a direct impact on lowering the consumption of energy. Extensive research on developing such materials is underway to improve the properties and performance of CFRTP for many industrial sectors to reduce corrosion effects, energy consumption, and overall manufacturing costs. Furthermore, CFRTP can be an alternative to other composite materials in blade production for wind energy applications. Prabhakaran et al. [146] and Mishnaevsky et al. [160] discussed the suitability of CFRTP and thermoplastic resin for future turbine blades and associated challenges for producing large blade structures from such materials. Additionally, the aerospace industry has recognized that CFRTP composite materials provide outstanding cost savings compared to traditional materials such as thermosetting composite materials and metal [25].

Despite these promising advantages of CFRTP, there are some drawbacks and challenges. One of the challenges is the high cost of virgin CF; although the cost dropped considerably in the past few years. Thus, the manufacturing cost of CFRTP is relatively high, however; it costs less than CF reinforced thermosetting polymers since it consumes less energy. These are some of the challenges that face the incorporation of the composite as a highly demanded material in many industries. Researchers have focused on developing recycling methods of CF and CFRTP composites, which are expected to lower the cost by 50% [161]. This will make using CFRTP material composites more economically sound in terms of the life cycle of the composites. Meanwhile, one of the main challenges in manufacturing CFRTP is that CF is non-polar while thermoplastics are polar materials. This disagreement in polarity results in poor interfacial adhesion of the manufactured composite unless the surface of CF was treated prior to the manufacturing process. Although the treatment process increases the manufacturing time and the cost of the end product, it is essential to ensure compatibility between CF and the polymer. In addition, polymers cannot withstand high temperatures and oxidation, unlike CF which must be taken into consideration when choosing a manufacturing process [22,23,24,25]. CFRTP are favorable composite materials due to their high strength-to-weight ratio compared to other conventional materials; however, it is rather difficult to estimate the fatigue of CFRTP, unlike metals. The lack of the fatigue (endurance) limit of CFRTP makes it challenging for engineers and designers to predict the exact fatigue failure properties of CFRTP. Also, CFRTP composites are hard, tough, and extremely abrasive which makes machining CFRTP a challenge. Without tools designed to withstand the damage CFRTP can cause, tool life can be very short when machining CFRTP composites. These challenges attracted many researchers into building trusted models that can accurately predict fatigue failure of the composite and achieve an ideal machining process [117,155,162].

Overall, CFRTP composite materials have become a progressively used class of lightweight materials. The research and development activities carried out to investigate the relationships between processing, structure, and properties of CFRTP have resulted in a better fundamental understanding of these materials and led to an enhancement of their properties, offering more flexibility in the design for several possibilities applications. Therefore, CFRTP is a promising candidate in a variety of industrial applications. The properties of CFRTP composite materials such as high strength, low weight, and good thermal and electrical properties make it a preferred composite material compared to neat polymers, CFRPs, and even other metallic materials. However, the polymer matrix and the treatment method of CF prior to the manufacturing process is crucial and will affect the composite properties; hence, it will also affect the applications of the composite material. Thus, the large-volume market applications of CFRTP are still to be discovered. Nevertheless, with the huge demand of emerging industries, the opportunities for improvements, and the support of developing standardizations for testing and using CFRTP composite, more high-efficiency CFRTP products will be developed.

## 4. Conclusions

In this comprehensive review, developments in the research on carbon fiber-reinforced polymers thermoplastic (CFRTP) have been explored extensively with a focus on the properties of the composite, such as the mechanical, electrical, and thermal properties. The outstanding properties exhibited by CFRTP are the primary motivation for further research and development. For example, these properties improved significantly with the addition of carbon fiber (CF) as a reinforcement compared to the neat polymer properties, which paves the way for CFRTP products in many industrial sectors. Furthermore, the modification of the CF’s surface is essential to improve the interfacial bond between the CF and the thermoplastic matrices. Either a chemical or physical modification technique will increase the oxygen concentration on the CF’s surface. Increasing oxygen makes the surface of CF more similar to the thermoplastic matrix in terms of polarity. Moreover, modifications have improved the filler/matrix bond and have had excellent positive effects on the mechanical properties of the composite compared to the untreated thermoplastic polymer/CF composites. A variety of modifying techniques for the surface of CF prior to the manufacturing process were discussed.

In general, the properties of various thermoplastic composites improved significantly with the addition of CF as a reinforcement compared to the neat thermoplastic properties. However, there is a variety in such improvements. This could be attributed to several factors, including manufacturing technique, processing parameters, thermoplastic type, CF type and orientation, loading, dimension, and surface treatment techniques, leading to interfacial adhesion and dispersion statues. All such aspects are essential to attain the anticipated properties, particularly mechanical properties, and to understand the relationships of the modification methods and mechanical properties of the final CFRTP composites. Therefore, this review provides the required stepping-stone to fully exploit the potential of CFRTPs in the manufacturing industry.

## Figures and Tables

**Figure 1 polymers-13-02474-f001:**
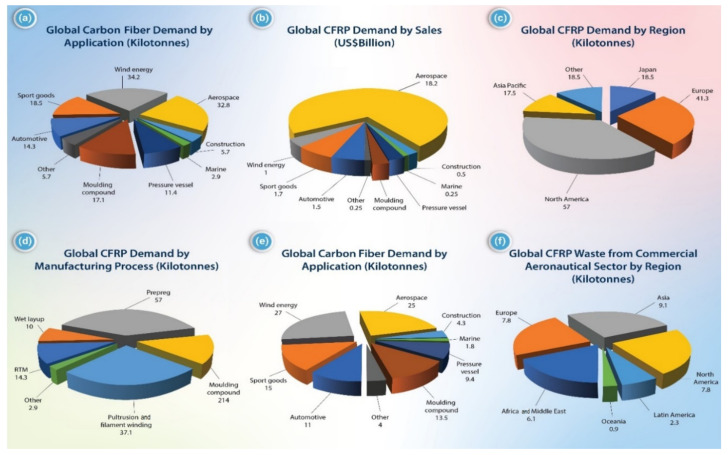
Global CFRP consumption in 2018 is categorized by (**a**) application, (**b**) sales, (**c**) region, and (**d**) manufacturing techniques. Global CF consumption in 2020 (**e**) by application and (**f**) estimated worldwide CFRP waste in 2050 from the aeronautical sector by region [6].

**Figure 2 polymers-13-02474-f002:**
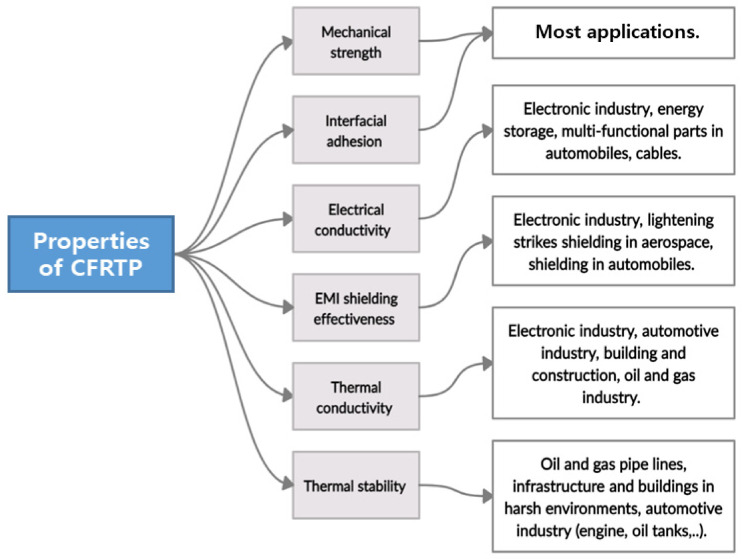
Properties and their connections to different industrial sectors.

**Figure 3 polymers-13-02474-f003:**
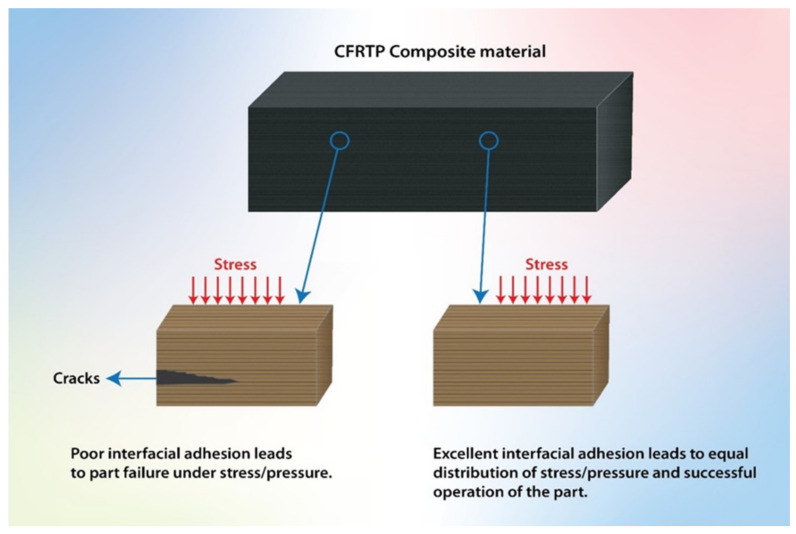
Schematic diagram of poor and excellent interfacial adhesion between the polymer matrix and the reinforcement.

**Figure 4 polymers-13-02474-f004:**
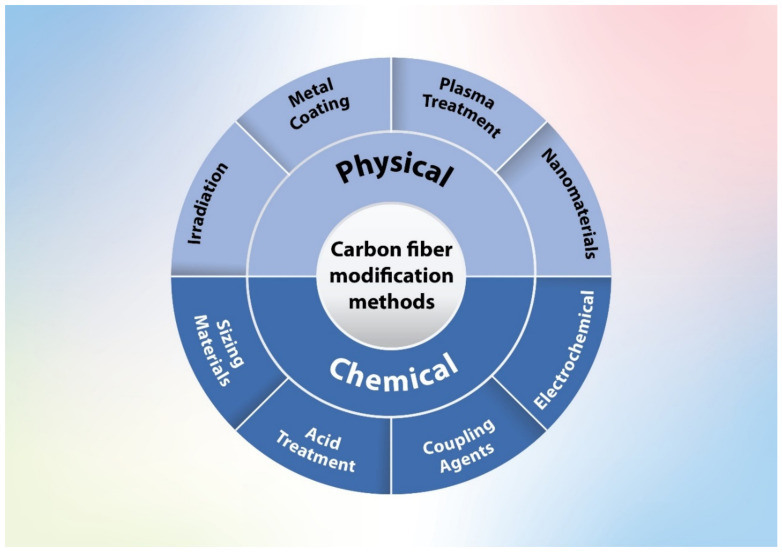
The standard treatment methods of CF surfaces.

**Table 1 polymers-13-02474-t001:** Types of thermoplastics include commodity and high-performance engineering plastics.

Thermoplastic Polymers					
Commodity Plastics			High Performance Engineering Plastics		
PE	PP	PS	PA	PEEK	PET
(C_2_H_4_)n Electrical high voltage applications in cables and water storage applications such as pipes and tanks.	(C_3_H_6_)n Manufacturing parts of automobiles, refrigerators, medical use, clothing, washing machines.	(C_8_H_8_)n Electrical and thermal insulation applications. Used frequently in the building and construction.	(CO-NH)n Used in manufacturing fibers and yarns as well as bearings, valves, and gears plus packaging industry.	(C_19_O_3_)n Used in manufacturing parts in automobiles and airplanes including gears, valves.	(C_10_H_8_O_4_)n Mostly used in food packaging, bottles, cloth, fibers, tapes, thermal and electrical insulations.

**Table 2 polymers-13-02474-t002:** The type of thermoplastic polymer and CF, modification techniques for CF, and mechanical properties of a variety of CFRTP composites.

Composite Material	CF Modification Method	Mechanical Properties	[Reference No.] (Publication Year)
PEI/CCF	Not mentioned	Young’s modulus, toughness, and % strain with respect to the loading direction were increased remarkably.	[15] (2010)
PP/RCF	Coupling agents followed by plasma treatments	An increase of up to 47.8% in ILSS.	[28] (2012)
CBT/CF	Microwave plasma treatments	Tensile strength enhanced by ~436.3%.	[34] (2014)
PP/SCF	Sizing materials followed by plasma treatment. A coupling agent was added to all samples	An increase of up to 47.8% in ILS.	[39] (2014)
PP/LCF	Coupling agent	The mechanical strength of the resulting composite was significantly enhanced.	[40] (2014)
PVDF/CF	Coupling agent	Flexural strength and modulus improved by 47% and 74%, respectively.	[44] (2016)
PA6/CF	Coupling agent and sizing treatment	The tensile strength of composites containing LCF was much higher than other composites having SCF by 24%.	[47] (2014)
PC/CF	Sizing materials	The tensile strength and modulus improved by 150% and 540%, respectively.	[48] (2014)
PPEK/CF	Sizing materials	The value of ILSS of sized CF is about 51.50 MPa, higher than the unsized CF, which was around 39.50 MPa.	[50] (2013)
PPEK/CF	Sizing materials	About 80% of the ILSS in PPEK/CF composite system was attributed to residual radial compressive stress at the fiber/matrix interface.	[51] (2013)
PC/shopped CF	Material coating followed by coupling agent treatment.	The strength and modulus of composites continuously increased along with a clear ductile-brittle transition by increasing the amount of CF.	[52] (2018)
HDPE/CF	Without using any coupling agent	Flexural properties increased significantly upon increasing the layer of CF in the composites.	[54] (2016)
POM/CF	Oxidation treatments	The flexural strength and modulus were enhanced remarkably.	[57] (2015)
PA12/CF	Oxidation method followed by coating with a layer of PA12	The flexural strength and modulus improved by 114% and 243.4%, respectively.	[60] (2011)
UHMWPE/CF	Acid treatment	A 70% increase in ILSS was observed.	[61] (2017)
PA6/ABS/SCF	Acid treatment	The tensile strength and tensile modulus improved significantly. However, these properties were enhanced dramatically when PA6 was blended with ABS.	[62] (2011)
PI/CF	Ozone modification and air-oxidation modification.	Improved friction and wear properties of the composite.	[64] (2010)
PP/RCF	Different plasma powers treatment	The tensile and flexural strength values of composites increased considerably by 17% and 11%, respectively, at 100 W.	[69] 2019
PP/CF	Irradiated PP as compatibilizer agent.	The tensile strength improved by 30%.	[70] (2013)
PP/SCF	Material coating	All flexural, tensile, and impact strength increased by about 43%.	[75] (2018)
PA6/CF	Material coating	ILSS increased by 40.2%.	[77] (2018)
PES/CF	Material coating	The maximum improvement was 12.1%, 31.7%, 12.4%, and 17.3% for the tensile strength, Young’s modulus, flexural strength, and flexural modulus, respectively.	[78] (2015)
PP/CF	Material coating	The ILSS increased by 300%.	[83] (2013)
PEEK/CF	Material coating	An increase of 115.4% and a 27% increase in impact toughness.	[84] (2017)
Nylon/CF	Not mentioned	An increase in ILSS of 33%.	[85] (2019)
ABS/CF	Not mentioned	An enhancement in hardness and compression strength was reported.	[87] (2014)
PPS/CF	Not mentioned	The strength, modulus, wear resistance, and hardness were improved significantly, although the strain values at break and impact strength were slightly decreased.	[88] (2013)
PTFE/CF & PPS/CF	Sizing materials	The strength, modulus, hardness and wear resistance, the elongation at break, and hardness were improved.	[89] (2016)
PTT/CF	Sizing materials	A significant tensile enhancement of up to 120% and flexural strength up to 30% were observed.	[91] (2012)
PA6/CF	Not mentioned	The addition of CF significantly enhanced flexural strength and modulus by 208% and 438%, respectively.	[92] (2014)
PA6/CF	Not mentioned	The results indicated that excellent tensile properties, including tensile modulus and strength and uniform CF distribution, have been proved.	[93] (2018)
PA6/CF	Not mentioned	An increasing CF loading led to improvements in tensile strength, modulus, and hardness, but reduced strain at break values of composites. Meanwhile, the investigated length ranges of CF (0 to 50 µm) had no effect on these mechanical properties except that strain at break was improved.	[98] (2013)
PP/SCF	Sizing materials	Tensile strength, tensile modulus, and impact strength were improved upon the increasing amount of SCF despite reducing fiber length, tensile strength, tensile modulus, and impact strength.	[102] (2020)
PBT/CF	Not mentioned	Improvements in tensile strength up to a certain amount of CF; however, further addition of CF led to a reduction in such property.	[103] (2016)
PI/CF	Not mentioned	The tensile strength was higher than 1200 MPa in the fiber direction on a temperature range varying from −50 to 250 °C but with low ILSS at high temperatures.	[104] (2016)
PEEK/CF	Not mentioned	Improvements in wear resistance were reported.	[105] (2019)
PEEK/CF	Coupling agent	The tensile strength and modulus increased by 455% and 168%, respectively.	[107] (2018)
PA6/CF	Coupling agent	The ultimate tensile strength, elastic modulus, and elongation at break values were exceptional.	[108] (2016)
PK/CF	Not mentioned	An enhancement in Young’s modulus of 520% and in tensile strength by 189%. In contrast, a significant decrease in the elongation at break was observed in the PK/CF composites even at very low loading.	[109] (2019)
PLA/SCF&CCF	Not mentioned	The tensile strength and flexural stress increased by 460% and 121%, respectively.	[113] (2021)
UHMWPE/CF	Not mentioned	Young’s modulus with the tensile strength significantly increased by 415% and 46%, correspondingly. However, the elongation/strain at break decreased substantially by 95%.	[114] (2014)

**Table 3 polymers-13-02474-t003:** The type of thermoplastic polymer and CF, modification techniques of CF, and electrical properties of a variety of CFRTP composites.

Composite Material	CF Modification Method	Electrical Properties	[Reference No.] (Publication Year)
PC/CF	Sizing materials	The highest electrical conductivity was around 0.0035 S/cm compared to about 0.0005 S/cm.	[48] (2014)
PP/SCF	Sizing materials	The electrical conductivity showed a strong dependence on the CF length. Two orders of magnitude in the electrical conductivity were reported.	[102] (2020)
PBT/CF	Not mentioned	The EMI SE improved significantly to around 10 and 11.16 dB for solid and foamed composites, respectively.	[103] (2016)
ABS/EVA/SCF	Coupling agent	A marginal increase in EMI SE with an increase in electromagnetic radiation frequency, but a sharp increase was observed with an increase in the SCF contents.	[125] (2011)
PP/CF	Not mentioned	The addition of CF resulted in a significant decrease in the percolation threshold value.	[128] (2014)
PP/CF with the addition of 1 wt% of CNTs	Material coating	A decrease in volume resistivity with an increase in the EMI SE was observed.	[133] (2014)
PP/CF	Not mentioned	The electrical percolation threshold was 8.75 and 7 vol.% for solid and foam composites, respectively. The dielectric permittivity improved, and the through-plane electrical conductivity increased by up to 6 orders of magnitude, resulting in an increase of 65% in these foamed composites’ specific EMI SE.	[136] (2013)

**Table 4 polymers-13-02474-t004:** The type of thermoplastic polymer and CF, modification techniques of CF, and thermal properties of different CFRTP composites.

Composite Material	CF Modification Method	Thermal Properties	[Reference No.] (Publication Year)
Polysulfone/CF	Not mentioned	The thermal conductivity was 1.82 W/mK at a CF loading of 26 vol.%.	[13] (2017)
PA6/LCF	Sizing materials	The in-plane and through-thickness thermal conductivity were 2.45 W/mK and 0.59 W/mK, respectively.	[46] (2013)
PEEK/CF	Sizing materials	Sized CF was more stable than untreated CF with an increase in surface energy and wetting performance.	[50] (2013)
PC/CF	Material coating followed by coupling agents	It was found that the lowest decomposition temperature was about 270 °C, and the highest temperature at which no further weight loss was observed was approximately 500 °C.	[52] (2018)
HDPE/CF	Without using any coupling agent	The thermal degradation temperature started at around 30 °C higher than that of the neat HDPE. Similarly, the maximum decomposition temperature was about 15 °C higher compared to pure HDPE. The multi-layered laminated composite panels’ thermal stability also improved by 41%.	[54] (2020)
PP/CF	Material coating	An improvement in thermal stability and crystallization temperature was reported.	[75] (2018)
nylon/CCF	Not mentioned	The nylon filament composites showed no melting peak attributed to the addition of CF to the thermal processing history of this polymer.	[85] (2019)
PTT/CF	Sizing materials	PTT composites containing CF exhibited an increase of more than 150 °C in the heat deflection temperature, and no significant changes in the melt temperature were observed.	[91] (2012)
PA6/CF	Not mentioned	The thermal conductivity at 80 °C was 0.21 W/mK, and increased by about 24% when the annealing temperature was increased to 190 °C. The thermal conductivity of composites increased to 0.32 W/mK and improved by 13% in the annealing process. The heat deflection temperature value for PA6 lies between 64 °C to 77 °C based on its annealing temperature; however, for the composites, it was around 214 °C with a negligible effect for thermal annealing. The results also showed that the T_g_ of PA6 increased from 60 °C to 78 °C with the incorporation of the CF.	[92] (2014)
PA6/CF	Not mentioned	The effect of SCF content and its length on the thermal properties of CF-reinforced PA6 composites. The results showed no change in the values T_g_ and T_m_ for the composites even with the CF loading increase. However, the heat of fusion and the degree of crystallinity of the composites decreased with the rise in the composites’ CF loading.	[98] (2013)
PI/CF	Not mentioned	The authors reported a longitudinal and transversal coefficient of thermal expansion of 1.7 × 10^−6^/°C and 30 × 10^−6^/°C, respectively, clearly depicting the materials’ high anisotropy. They reported that above 200 °C, the expansion with an increase in temperature was non-linear. Two transitions were observed at approximately 235 °C and 385 °C, attributed to Tg and Tm of the polymeric material, respectively.	[104] (2016)
PEEK/CF	Coupling agent	The degradation started at around 408 °C, and the decomposition maxima occurred at 468 °C, and the maximum rate of degradation was 2.3%/min. Improvement by 20 °C in the initial degradation temperature was observed.	[107] (2018)
PK/CF	Not mentioned	An enhancement of thermal conductivity up to 300%.	[109] (2019)
CBT/SCF	Sizing materials	The T_g_ increased by 19.5%. Overall, thermal degradation of improved for all investigated composites compared to plain PP.	[110] (2016)
ABS/EVA/SCF	Coupling agent	An enhancement in thermal stability was reported.	[125] (2011)
PP/CF	Not mentioned	No change in the melting behavior was reported.	[139] (2012)
PI/CF	Not mentioned	An improvement in the heat distortion temperature by 10 °C. T_g_ and T_m_ showed no considerable changes. The thermal conductivity improved from 0.17 W/(mK) to 0.49 W/(mK) in the molding direction and 0.22 W/(mK) in the transverse direction.	[140] (2010)
PI/CF	Not mentioned	The T_g_ value obtained showed an increase to 241 °C and 244 °C.	[141] (2018)
PBT/PES/CF	Oxidation treatment followed by material coating	Their result showed improvement in T_g_ from 44 °C to 46 °C for composites containing PBT. However, the T_g_ of composites containing PES decreased from 218 °C to 212 °C.	[142] (2019)

## Data Availability

Not applicable.

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
