# Peer review of "Comprehensive Review of the Properties and Modifications of Carbon Fiber-Reinforced Thermoplastic Composites"

_polymers, 2021, doi:10.3390/polym13152474_

Round 1

Reviewer 1 Report

After going through the manuscript " Comprehensive Review on Properties and Modifications of Carbon Fiber Reinforced Thermoplastic Composites", I would give my comments below.

The manuscript is well organized; however, to improve the quality, the following recommendations can be incorporated.

- I think it's a parallel work with some new review papers that publish in recent months such as:

“Khurshid, M. F., Hengstermann, M., Hasan, M. M. B., Abdkader, A., & Cherif, C. (2020). Recent developments in the processing of waste carbon fibre for thermoplastic composites–A review. Journal of Composite Materials, 54(14), 1925-1944..“

so, what makes this review different from the others and from the most recent ones?

- Prepare statistical data (such as the number of documents, document per country) about you used references by created databank such as Scopus, Google scholar, and web of science.

- There are some grammatical errors, please carefully check the whole manuscript.

- A review paper not only should summarize recently published works, but also should contain critical and comprehensive discussions. Therefore, check writing for the whole manuscript. The review should not be presented by listing what have done by others.

Reviewer 2 Report

This is a general review of carbon-fibre reinforced polymer composites. I am missing though a couple of elements:

What about hybrid systems? There are just two related references, 51 and 114; but for sure there are more. Generally, it would be good to define the boundaries and search strategy of the review in the beginning.

I feel that the "future prospectives" section (3) is disproportionally small compared to the rest of the review. I think that discussion on the material aspects that render CF beneficial (or not) should be extended, rather than focusing on the collection of results and future possible market trends.

Reviewer 3 Report

This review summarizes Properties and Modifications of Carbon Fiber Reinforced Thermoplastic Composites. This topic is not entirely new, however this information is of interest to the public. In the following, the comments are given to improve the weak points of the paper.

There are 147 references in the paper given in total. However, a large part of the references do not come from current publications. There are only 3 references from 2021, 6 references from 2020, 13 references from 2019 and 2018. More current references from 2018-2021 must be included.

All sources in the list of references should be checked and formatted consistently according to the MDPI template. In particular, Internet sources should be revised (references 146 and 147). In addition, reference 1 is not presented correctly. Reference 20 should be written without capital letters and the indication of journal etc. is missing. In addition, not all references include DOI.

Furthermore, the authors should check and delete unnecessary empty spaces in the paper.

In addition, the references in the text are not always consistent and not given in order and jump in some parts of the paper.  For example, in line 698 the reference 132 comes first and then directly 48. Or in line 701 the reference 133 is given and then in line 707 the reference 102 suddenly appears. This is repeated in several parts of the paper. It makes the impression that the paper was written chaotically and the references were put together randomly.

Line 21: Authors describe in abstract part that carbon fiber reinforced thermoplastic polymers exhibit low manufacturing costs. This is not entirely true because the production of CF requires a lot of energy during the oxidative stabilization and subsequent carbonization processes and cannot exhibit low manufacturing costs. This statement should be revised.

Line 40: A lot of information is given in the paper and partly without references, for example line 40 to 43 or only after a long passage with several sentences. The authors should check such passages in the whole paper and, if necessary, insert the sources directly at the end of the sentence where this information comes from. Another bad example is line 387 with 7 sources at the end of the sentence [3,7,17,35,56,63,69,71]. Better to give these references at the end of the sentence where this information comes from.

Line 44: In Introduction part, carbon fibers precursors such as rayon, or petroleum pitch are mentioned. However, the most important precursor - polyacrylonitrile (PAN) - is missing.

Line 47: "It is well known that higher carbonization temperatures can achieve high carbon content in CF." This is a very general statement. At this point the temperature range should be given at which carbonization stage takes place, because at higher temperatures graphitization stage occurs. It should also be mentioned why the production of carbon fibers takes place under stretching and what effect is achieved.

Line 71: At this point it should be briefly mentioned that the matrix is polymer resin for the embedding of reinforcing fibers.

Line 94: Please change the word to matrix instead of matrices. It is used more often.

Figure 2: The graphic is deformed and should be corrected.

Line 132: One word should be changed here to "In conclusion, this report offers a comprehensive overview of..."

In general, the abbreviations in the whole paper should be checked and written out for a better understanding by the reader (see below).

Line 150: what is "EMI shielding"?

Line 204: PC/CF

Line 209: PVDF

Line 219: Please check: (PA) polyimide

Line 227: PI

Line 233: O/C ratio

Line 245: PET

Line 251: HDPE

Line 259: UHMWPE

Line 265: POM

Line 267: PS

Line 287: ABS

Line 331: HDPE

Line 391: ILSS

Line 638 and 640: Why is "EMI shielding effectiveness" emphasized and made bold here?

Line 650: Please check the formatting.

Tables 1, 2 and 3 are not well presented, chaotically done and do not give clear information and therefore why are these tables presented. These tables should be arranged more meaningfully and the information especially in the first and third column should be given more consistently. Partly there are only remnants of the information and it is not clear why it is done. At this point, the authors need to organize the tables more clearly, reduce unnecessary sources and present the information more uniformly and precisely. In particular, the first column should be sorted by material types. Currently, for example, in table 1, PP is mentioned in the first column, followed by PA6, PS and then PP again. It seems that the authors have carelessly taken the materials and there is no clear line why it was ordered in this way. 

In addition, sometimes the temperature signs are different as in Table 3, source 104.

Line 904: Future prospectives is not clearly presented and should be revised.

Line 911: The source of statistical data for the year 2029 is missing.

Round 2

Reviewer 1 Report

The manuscript needs to "drawbacks and challenges" section. In addition, the language of the manuscript should be improved.

Author Response

Reviewer 1

The manuscript needs to "drawbacks and challenges" section. In addition, the language of the manuscript should be improved.

Thank you for this useful comments. The drawbacks and challenges of CFRTP were added in the future prospective section. Also, the language of the manuscript was edited by MDPI editing service.

Reviewer 2 Report

The authors have addressed the comments of the referees well and the paper can be published now. (Minor spelling check is needed during publication).

Author Response

Reviewer 2

The authors have addressed the comments of the referees well and the paper can be published now. (Minor spelling check is needed during publication).

Thank you for the kind comment. The language of the manuscript was edited by MDPI editing service.

Reviewer 3 Report

The authors have revised the manuscript well according to the suggestions of the reviewers and after minor revisions the paper can be recommended for publication.  Enclosed are some comments for minor improvement of the manuscript:

In general, a native speaker should review the manuscript and spelling mistakes and unnecessary blanks should be revised. It is of course possible that there are blanks in various places due to the different WORD versions and conversion to PDF. However, authors should carefully check and revise everything in the final manuscript in order to avoid these minor issues.

Table 2 - Improvments (reference 103) - the word is misspelled

Table 2, 3 and 4- [Reference No.] (Puplication year ) - the word is misspelled

Table 4 - Source 104 - wrong temperature signs at 200°C

Last line in table 4 - different size of letters and temperature indication is not correct (PBT/PES/CF)

Line 647 - still wrong style of letters

Table 4 - An enhamcment in thermal stability was reported (Source 125) - the word is misspelled

Table 4- missing heading

All tables contain small errors and sometimes the letters are the wrong size or start with lower case letters. The authors need to improve the tables.

The authors should check all references carefully and format them consistently according to the Journal template. The references are still very poorly and differently formatted and not according to template. Currently there are different formats and styles. Some references are missing volume or page number, etc. There should always be three things, year, volume and page number.

Reference 1 - incorrectly formatted and some details such as volume are missing.

Park S.J., Heo GY. Precursors and Manufacturing of Carbon Fibers. In: Carbon Fibers. Springer Series in Materials Science, 2015, vol 210. Springer, Dordrecht. https://doi.org/10.1007/978-94-017-9478-7_2

Reference 4 - missing page number (1667)

Reference 6- missing page number (108053), the figure of 1018 is wrong.

Reference 20 - missing - J. Mech. Manuf. 2018, 1, 144-154.

Reference 27- please find another but a scientific source.

Reference 33 - missing page number (651)

The following references are also missing page numbers etc. and should be checked: 36, 79, 87 (space), 109, 113, 145 (space), 144, 146, 147 - formatted incorrectly, 148, 151,153, 154, 157, 158

Reference 138 - is formatted incorrectly. Correct would be:

Yajima, S. Silicon carbide fibres, in Watt, W. and Perov, B.V. eds. Strong fibres, Handbook of Composites, series eds. A. Kelly and Y.N. Rabotnov, North-Holland Amsterdam, 1985; 1; 201-240.

Reference 144 - is misrepresented. Correct would be:

Burela, R. G., Kamineni, J. N., & Harursampath, D. Chapter 8 - Multifunctional polymer composites for 3D and 4D printing, Editor(s): Kishor Kumar Sadasivuni, Kalim Deshmukh, Mariam Alali Almaadeed, 3D and 4D Printing of Polymer Nanocomposite Materials, Elsevier, 2020, 231-257, https://doi.org/10.1016/B978-0-12-816805-9.00008-9

Author Response

Reviewer 3

The authors have revised the manuscript well according to the suggestions of the reviewers and after minor revisions the paper can be recommended for publication.  Enclosed are some comments for minor improvement of the manuscript:

In general, a native speaker should review the manuscript and spelling mistakes and unnecessary blanks should be revised. It is of course possible that there are blanks in various places due to the different WORD versions and conversion to PDF. However, authors should carefully check and revise everything in the final manuscript in order to avoid these minor issues.

Thank you for the wonderful comments. The language of the manuscript was edited by MDPI editing service.

Table 2 - Improvements (reference 103) - the word is misspelled

Table 2, 3 and 4- [Reference No.] (Publication year) - the word is misspelled

Table 4 - Source 104 - wrong temperature signs at 200°C

Last line in table 4 - different size of letters and temperature indication is not correct (PBT/PES/CF)

Line 647 - still wrong style of letters

Table 4 - An enhamcment in thermal stability was reported (Source 125) - the word is misspelled

Table 4- missing heading

All tables contain small errors and sometimes the letters are the wrong size or start with lower case letters. The authors need to improve the tables.

All misspelled words were corrected and whole manuscript edited by MDPI editing service.

The authors should check all references carefully and format them consistently according to the Journal template. The references are still very poorly and differently formatted and not according to template. Currently there are different formats and styles. Some references are missing volume or page number, etc. There should always be three things, year, volume and page number.

Reference 1 - incorrectly formatted and some details such as volume are missing.

Park S.J., Heo GY. Precursors and Manufacturing of Carbon Fibers. In: Carbon Fibers. Springer Series in Materials Science, 2015, vol 210. Springer, Dordrecht. https://doi.org/10.1007/978-94-017-9478-7_2

Reference 4 - missing page number (1667)

Reference 6- missing page number (108053), the figure of 1018 is wrong.

Reference 20 - missing - J. Mech. Manuf. 2018, 1, 144-154.

Reference 27- please find another but a scientific source.

Reference 33 - missing page number (651)

The following references are also missing page numbers etc. and should be checked: 36, 79, 87 (space), 109, 113, 145 (space), 144, 146, 147 - formatted incorrectly, 148, 151,153, 154, 157, 158

Reference 138 - is formatted incorrectly. Correct would be:

Yajima, S. Silicon carbide fibres, in Watt, W. and Perov, B.V. eds. Strong fibres, Handbook of Composites, series eds. A. Kelly and Y.N. Rabotnov, North-Holland Amsterdam, 1985; 1; 201-240.

Reference 144 - is misrepresented. Correct would be:

Burela, R. G., Kamineni, J. N., & Harursampath, D. Chapter 8 - Multifunctional polymer composites for 3D and 4D printing, Editor(s): Kishor Kumar Sadasivuni, Kalim Deshmukh, Mariam Alali Almaadeed, 3D and 4D Printing of Polymer Nanocomposite Materials, Elsevier, 2020, 231-257, https://doi.org/10.1016/B978-0-12-816805-9.00008-9

All references were checked carefully and all your remarks have been addressed. Thank you for your valuable review.